# A descriptive study of zoonotic disease risk at the human-wildlife interface in a biodiversity hot spot in South Western Uganda

**Shamilah Namusisi**[1]*, **Michael Mahero**[2], **Dominic Travis**[2], **Katherine Pelican**[2☉], **Cheryl Robertson**[3☉], **Lawrence Mugisha**[2,4,5☉]

**1** School of Public Health, Makerere University, Kampala, Uganda, **2** Department of Veterinary Population Medicine, University of Minnesota, Saint Paul, Minnesota United States of America, **3** School of Nursing, University of Minnesota, Minneapolis, Minnesota, United States of America, **4** Ecohealth Research Group, Conservation & Ecosystem Health Alliance (CEHA), Kampala, Uganda, **5** College of Veterinary Medicine, Animal Resources & Biosecurity (COVAB), Makerere University, Kampala, Uganda

☉ These authors contributed equally to this work.
* shamilahnamusisi@gmail.com

**Data Availability Statement:** All relevant data are within the manuscript.

## Abstract

Zoonotic diseases pose a significant health challenge at the human–wildlife interface, especially in Sub-Saharan Africa where ecosystem services contribute significantly to local livelihoods and individual well-being. In Uganda, the fragmented forests of Hoima district, form part of a "biodiversity and emerging infectious disease hotspot" composed of communities with high dependency on these wildlife protected areas, unaware of the associated health risks. We conducted a cross-sectional mixed methods study from March to May 2017 and interviewed 370 respondents, using a semi-structured questionnaire from eight villages neighbouring forest fragments in Hoima District, Uganda. Additionally, a total of ten (10) focus group discussions (FGDs) consisting of 6–10 men or women were conducted to further explore the drivers of hunting and perception of zoonotic disease risks at community level. Qualitative and quantitative data were analysed using content analysis and STATA version 12 respectively. We found twenty-nine percent (29.0%, CI: 24.4–33.9) of respondents were engaged in hunting of wildlife such as chimpanzee (*Pan troglodytes)* and 45.8% (CI: 40.6–51.0), cane rats (*Thryonomyidae spp*). Acquisition of animal protein was among the main reasons why communities hunt (55.3%, CI: 50.1–60.4), followed by "cultural" and "medicinal" uses of wildlife and or its parts (22.7%, CI: 18.6–27.4). Results further revealed that hunting and bushmeat consumption is persistent for other perceived reasons like; bushmeat strengthens the body, helps mothers recover faster after delivery, boosts one's immunity and hunting is exercise for the body. However, respondents reported falling sick after consumption of bushmeat at least once (7.9%, CI: 5.3–11.1), with 5.3% (CI: 2.60–9.60) reporting similar symptoms among some family members. Generally, few respondents (37.0%, CI: 32.1–42.2) were aware of diseases transmissible from wildlife to humans, although 88.7% (CI: 85.0–92.0) had heard of Ebola or Marburg without context. Hunting non-human primate poses a health risk compared to edible rats (cane rats) and wild ruminants (cOR = 0.4, 95% CI = 0.1–0.9) and (cOR = 0.7, 95% CI = 0.2–2.1) respectively. Study suggests some of the pathways for zoonotic disease spillover to humans exist at interface

**Funding:** This research was supported by NIH Research Training Grant # D43 TW009345 funded by the Fogarty International Center through University of Minnesota. Additional funding was also obtained from the Department of Veterinary Population Medicine, College of Veterinary Medicine, University of Minnesota to enable completion of the work. The funders had no role in study design, data collection and analysis, decision to publish, or preparation of the manuscript.

**Competing interests:** The authors have declared that no competing interests exist.

areas driven by livelihoods, nutrition and cultural needs. This study offers opportunities for a comprehensive risk communication and health education strategy for communities living at the interface of wildlife and human interactions.

## Author summary

Zoonotic diseases are increasingly becoming an emerging public health threat, partially due to the risk of spillover events at the human-wildlife interface. This cross -sectional study was carried out among high-risk communities around forest fragments of Hoima District, Uganda, to describe activities that could lead to disease spillover in humans in order to inform public health practitioners of the potential risks at community level for preparedness and response efforts. The study also sought to gain an understanding of the community's perception of the risk of zoonotic diseases, what activities could expose them to zoonoses, and whether the healthcare services are adequate to identify such diseases at community level. We found that most people were not aware of zoonotic diseases transmissible from wildlife to humans and this can partly be explained by lack of information filtering through to the grass root. It is important to note however, that the interaction between wildlife and humans, is largely driven by communities' struggle to survive and meet their livelihood needs making it difficult to predict under what circumstances disease could emerge in the community. We need to remind ourselves that all major outbreaks have started at community level and well as the health experts are fast to diagnose the disease in question, some communities are hearing these diseases for the very first time. Quite often these outbreaks are putting the available alternative livelihood in question and demanding for quick unstainable changes that communities are quick to abandon once the outbreaks are over. We recommend that this is the time to invest in health education and create awareness about zoonotic diseases among communities at the human-wildlife interface. Health promotion and or livelihood-based intervention programs should use existing evidence and case studies implemented in collaboration with government agencies and partners.

## Introduction

Wildlife are known to be common reservoirs for some infectious diseases transmissible to humans [1]. It is estimated that more than 60% of infectious diseases in humans are of zoonotic origin causing a billion cases of illness and millions of deaths every year [2]. According to the International Union for the Conservation of Nature (2005), human-wildlife interaction is increasing due to human choices like land use or the need for ecosystem services that proximity to natural resources provide [3,4]. The burden of infectious diseases is noticeably high in Sub-Saharan Africa [5–7]. In addition, poor communities are disproportionally affected by climate and environmental changes that further drive the emergence of infectious diseases [8,9].

Uganda's vulnerability to climate change has been highlighted and is.bound to increase because many livelihood are dependent on natural resources [10]. Hoima District in Uganda is located close to the Congolese border in South West Uganda between two major forest blocks (Bugoma and Budongo) within the "biodiversity and emerging infectious disease hotspot" of the Albertine Rift Region [11]. Additionally, this area forms a mosaic of agricultural land, forest, woodland and grassland [12]. The forest fragments are faced with the increasing challenge of unregulated timber extraction and clearance for agriculture [13]. The human

population (majorly comprised of the *Bunyoro*, *Bakiga* and *Lugbara* tribes) resident in this area commonly live close to forest fragments—often less than 1000ha in size and within 1 km of a forest edge [12,14]. These fragments are inhabited by a mobile population of about 5000 chimpanzees *(Pan troglodytes schweinfurthii)* that move within and between forest-farm habitats, causing increased conflict with human farming communities [12]. The other wildlife present include: black-and-white colobus (*Colobus guereza occidentalis*), vervet monkeys (*Chlorocebus aethiops)*, tantalus monkey (*Chlorocebus tantalus budetti*), blue monkey (*Cercopithecus mitis stuhlmanni*), red-tailed monkey (*Cercopithecus ascanius schmidti*) and olive baboon (*Papio anubis*), the gray-cheeked mangabey (*Lophocebus albigena johnstoni)*, buffalo (*Syncerus caffer)*, giant forest hog (*Hylochoerus meinertzhageni)*, hippopotamus (*Hippopotamus amphibius*),spotted hyaena (*Crocuta crocuta*), leopard (*Panthera pardus*), Rwenzori duiker (*Cephalo phus rubidus*), topi (*Damaliscus lunatus*), cane rats (*Thryonomyidae spp)*, squirrels, and porcupine (*Hystrix cristat*a) [12].

There is a close link between emerging infectious disease spillover to humans and deforestation, biodiversity loss and forest invasions [15]. The intimate and dynamic human-wildlife interface in Hoima District, Uganda, sets the stage for infectious disease emergence from wildlife. It is important to note that since 2000 Uganda has suffered a total of 16 hemorrhagic fever outbreaks including Ebola virus (EBOV), Marburg, Crimean Congo and Rift Valley [16]; with current threat of local emergence or importation of EBOV from the ongoing outbreak in the Democratic Republic of Congo(DRC) [17]. This interface is particularly important where communities are less aware of the consequences of their activities and where public health systems are less developed [18,19]. Thus, characterizing and managing this threat poses both social and institutional challenges, emphasizing the need for effective multi-sectoral collaboration to enhance health surveillance and response systems supported by strong educational and policy frameworks [8,20,21]. While it is recognised that initial identification of emerging infectious zoonotic disease outbreaks have mostly occurred at the community level [22], there is little published data adequately describing the risk of disease emergence with respect to these communities [23]. Furthermore, the collection of ethnographic data in this area may shed light on potentially risky behaviors and activities, as well as local perceptions of risk, to better describe the overall risk of zoonotic disease emergence [20,24].

Human activities like encroachment on wildlife habitat for agriculture purpose has been highlighted as a risk of zoonotic transmission from wildlife to humans, for example, degradation that may result in higher contact within existing habitat or cause significant migration of wildlife out of the degraded environments into human settlements [11]. Additionally, several studies have classified these activities as high risk for zoonotic disease transmission to humans [15,25]. However, there is need to recognise that communities are pressed with survival needs and are often less aware of infectious zoonotic disease risks that could adversely affect their health and, if they do, the need for survival often outweighs the risk of infection. Therefore, the goals of this study were to: a) describe the nature of human-wildlife interaction that occur among communities around the forest fragments of Hoima, b) identify the potential pathways for disease spillover to humans from wildlife interaction, c) provide a synopsis of the health care services available and d) come up with some recommendation for interventions that will reduce such risks and the burden of these zoonoses among high human- wildlife contact communities.

## Methodology

### Ethics statement

All participants provided informed consent. Ethics approval was obtained from School of Biomedical Sciences Institutional Review Board of Makerere University (SB-HDREC-412),

University of Minnesota (STUDY00000469) and research permit issued by Uganda National Council of Science and Technology (UNCST HS2200). Written and oral informed consent was obtained from all participants. In some cases, willing participants were unable to sign because they were unable to read and write in which case oral consent was obtained. Oral consent was documented by; recording date and time of the interview on the consent form, indicating on the consent form "participant consented orally" in the space where the participant would have signed, interviewer and witness signing the consent form in their space. Both consent processes were approved by IRB and UNCST. Furthermore, local community approval was obtained from Hoima district local government and local village leaders.

## Study area

The study was carried out in the sub counties of Kitoba and Kiziranfumbi. Kitoba sub-county has a total 34,810 human population, land area of 195.5 square kilometers and population density of 344 persons per square kilometer of land area. Kiziranfumbi has a population of 35,814, land area of 223.3 square kilometers and population density of 345 persons per square kilometer of land area. Overall, Hoima is ranked among the top ten most heavily populated districts of Uganda (population 572,986), of which 81.5% live in rural areas (Uganda, UBOS 2014).

## Sample frame

To infer potential risk of exposure to wildlife-associated zoonotic diseases, a cross-sectional ethnographic mixed-method study was undertaken to describe the interface between wildlife and people in and around forest fragments in Hoima district. Data were collected from eight rural communities (villages) located within 1-2km of forest fragments in two sub-counties (Kitoba and Kiziranfumbi) (Fig 1) located between Bungoma and Budongo forests in Hoima District, Uganda. The selection of forest fragments was based upon highest likelihood of contact with wildlife, and included factors such as loss of tree cover (habitat disturbance), presence of wildlife in the community, high incidence of reported human-wildlife conflict and accessibility to the area.

Within fragments, villages were selected based on reports from the local vermin control office, which represent a higher level of human-wildlife contact or conflict (ranging from crop raids to human attacks). A semi-structured questionnaire was administered using face-to-face interviews to 370 participants (46 respondents per village) from eight villages. Within villages, participants were randomly selected with the goal of a male: female ratio of approximately 1:1. The number of total participants was derived using a list of household provided by the local leader and using formula = RANDBETWEEN(1,60)) to generate household random numbers in Microsoft Excel version 2016. To further explore communities perception on human-wildlife interaction, a total of ten (10) focus group discussions (FGDs) consisting of 6–10 men or women were conducted. Inclusion criteria for focus group participation included: use of forest services like hunting, fetching of firewood, water, and agriculture and timber harvest, willingness to participate and resident of the village.

## Qualitative data collection

Focus group discussions were conducted in a local language (*Runyoro)* employing the services of a local translator who also acted as a moderator. The principal investigator (first author) participated in all sessions and was responsible for all data collection (notes and voice recordings). Focus group discussions were guided by a four-part guide developed based on information collected during author's attendance at community meetings and informal interview and

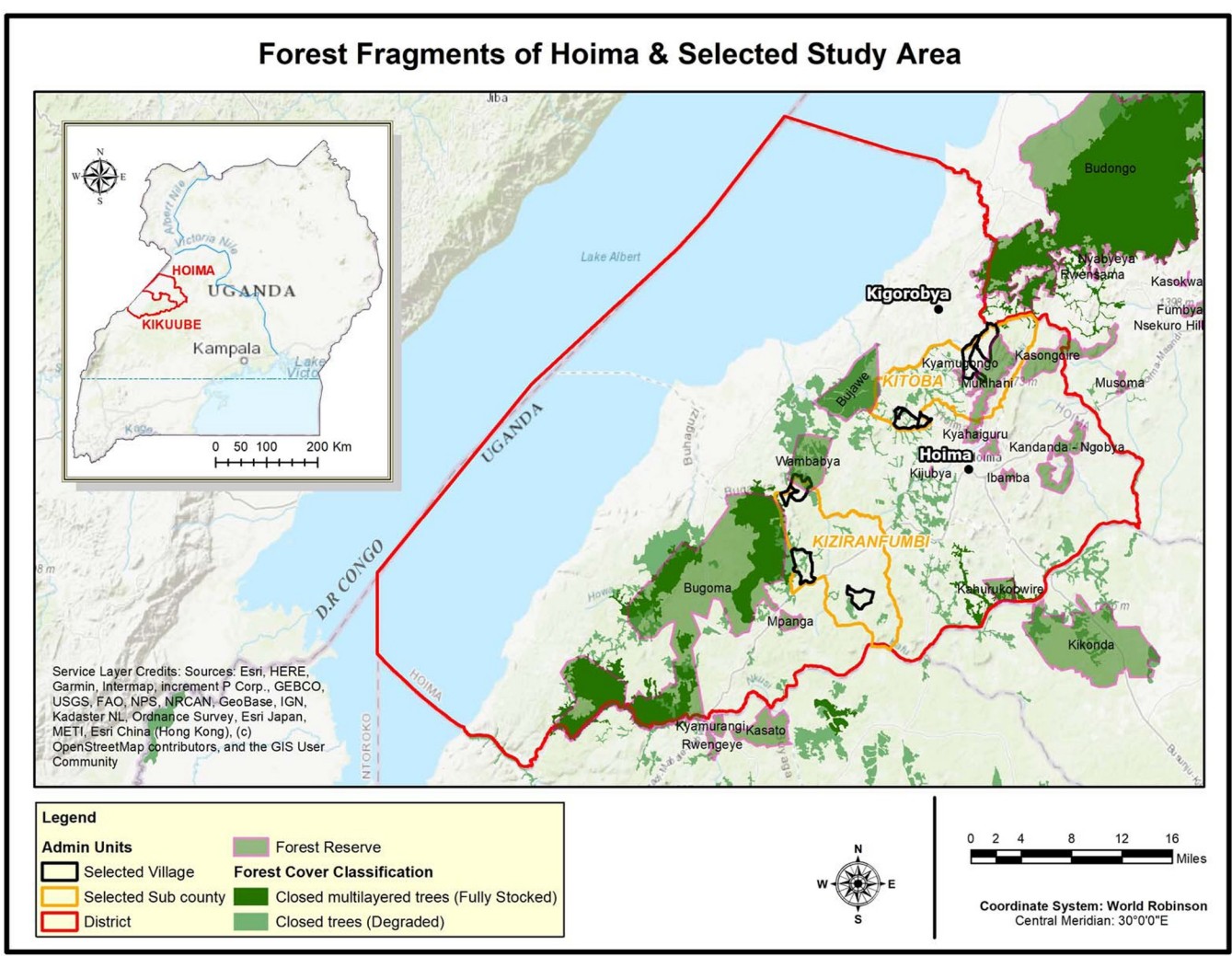

**Fig 1. Map showing study areas. Source:** Developing an experimental methodology for testing the effectiveness of payment for Ecosystem services to enhance conservation in production landscapes in Uganda (UNEP, 2017). Maps were generated using Quantum GIS 2009 package.

discussions. The four parts include content and context, human-wildlife interaction and health risks, illness in humans, and health seeking behavior and practices.

## Quantitative data collection

Quantitative data was collected using services of trained interviewers using face to face semi-structured questionnaire that included information on socio-demographic factors, activities that characterize human-wildlife interaction, knowledge and practices regarding zoonotic disease and existence/use of healthcare services (availability of health centers within 15km).

## Qualitative data analysis

The translator transcribed recordings to word transcripts in Microsoft Word. This was checked by employing a second person from the team of interviewers who together with the first author, listened to the recordings while reading the word transcripts to ensure that that transcripts were a true recording of the FGDs Transcripts were analyzed using content analysis

by coding, categorizing of related codes and identification of themes. To ensure that coding and resulting patterns reflected the experience of the interviewees, the first author together with translator reviewed the coded documents in Microsoft Word. Validity of the data was established through maintaining an audit trail of audiotapes, field notes, participant observation, translator making a follow up call to an interviewee identified during FGD session and use of analysis memos written during the coding process. The unit of analysis was the group and units of coding were phrases, sentences or paragraphs. Two broad themes related to the aim of the study emerged "human-wildlife interaction" and "health".

## Quantitative data analysis

Quantitative survey data was collected and cleaned using EpiData 3.0 and analyzed using Stata version 12 (StataCorp LP). Descriptive statistics on demographic characteristics of respondents were summarized using frequencies and percentages. Pearson chi-square test was used to find if there was a significant association among the independent variables taking the location as dependent variable and their significance levels at 95% level of confidence. Binary logistic regression was used to analyze the factors associated with illness after consumption of bush meat. At bivariate level, crude odds ratios (cOR) and their p-values with a statistical significance of 0.05 are presented to measure the association between each independent and dependent variable. The independent variables included Sex, age groups, household size, knowledge of diseases transmitted by wild animals to human, ever heard of Ebola or Marburg, mostly frequently hunted animals and Number of people involved in preparation of hunted animals. The dependent variable was defined as a person feeling unwell after consumption of bushmeat, transmitting a similar condition to other household members and or presenting himself or herself for medical care after consumption of bush meat.

At multivariate level, independent variables with p-values less than 0.5 at bi-variate level were included in the regression model and after adjusting for confounding, adjusted odds, ratios (aOR) and their p-values are presented. Using forward selection of variables, model was built using a step-wise procedure for selecting independent variables by adding and removing variables from the model until a good simple model was obtained. The model fit was assessed using log-likelihood chi-square and the stepwise model selection was done manually.

## Results

### Qualitative results

Two major themes emerged from the focus group discussions; human-wildlife interaction and health. Increased interaction of humans with wildlife occurs (content and context) within boundaries of several livelihood activities and this includes encroachment on forest reserves for crop farming and human settlement. Settlement around forest patches was found to be motivated by easily accessible resources like firewood, local weaving material, hunting and water. Wildlife on the other hand, also infringe upon human settlement to forage and through sharing of open water sources (rivers and wells) and fruits, providing an intersection where both humans and wild animals often interact. Occasionally there are attacks on humans by wildlife at these intersections.

> "We share fruits like mangoes with monkeys. We even find them on the road and at wells. Snakes bite people. Baboons bit someone and they escaped. I think the person annoyed them or something like that. Wild pig bit me 7 years ago and the site itches me up to now. A chimpanzee slapped someone. Am telling you we have lived and have suffered with these animals, but this is where we live" (Women's' FGD Kaigo)

## Hunting and consumption of bushmeat

Most commonly hunted and consumed wild animals in the community included edible cane rats (*Thryonomyidae spp.*), wild pigs and wild ruminants (bushbucks and antelopes) prepared and eaten in different forms. This meat is available in the community and one can find it whenever need arises,

*The animals we eat are cane rats (emisu), bushbucks (ensa) and other antelopes. They bring roasted meat and you may never know what it is. We get wild meat from hunters. They bring it in and say this is such and such meat. When it is roasted, you eat straight away. Sometimes we try to put in groundnuts. They sometimes bring fresh meat. There is a man who always brings in fresh antelope meat. It is common and there are people who eat it daily. It is common if you have money and are interested you can't take a week without eating it. It is just lovely; it is more delicious and increases after preparing (Women's FGD Kibanjwa).*

However, other wildlife types in addition to those consumed, are also hunted for various reasons contributing to persistent hunting behavior in these communities. Participants described different ways wildlife and wildlife parts are used as shown in Table 1.

Some communities had to say why they continued hunting

*" In hunting, we give security for our gardens. When you eat game meat you gain your immunity. In hunting you even do exercise". (Men's FGD Rwemisaga)*

## Disease transmission from wild animals: Are people aware of health risks involved?

Most respondents discussed that there were strange diseases, with fever and diarrhea as the main symptoms in the community. The majority could not explain whether these diseases were related to close contact with wildlife; in part, due to the fact that there is no linkage between health testing services and wildlife interaction at health facilities *"When one goes and is tested, they will not tell you that you have been found with an illness from animals".*

In some cases, people mentioned that diseases like Ebola, Brucellosis, Tuberculosis (TB) and Rabies are spread from animals in rather unsure ways. *"We hear Ebola comes from monkeys and baboons. Even eating dead animals–not knowing the cause of death" (Men's FGD Rwamisaga).* Sentiments shared by one participant reveal individuals experience with some of these zoonotic illnesses.

*"I have suffered Brucellosis; another woman went as far as Kampala due to Brucellosis and jiggers from pigs. People can also get rabies say when you are bitten by squirrels, eating half cooked pork, eating monkeys all have been said that can cause disease although we have not*

**Table 1. Representative examples on use of wildlife and wildlife parts in the community as described by participants.**

| Wild animal | Quotes from FGDs |
| --- | --- |
| Edible cane rats (*Thryonomyidae spp.*) | *The skull is prepared for children to cure measles. The feet, hairand teeth are medicine. The hair is oxytocic (quickens labor).* |
| Colobus monkeys (*Colobus guereza*) | *People also love tails and skins, say in church you see some one swing calabash tails, in cars and at ceilings. Skins are used by dancers. Hair also helps treat nose bleeding.* |
| Porcupine (*Hystrix cristata*). | *Porcupine pins [are used]to prick on swollen breasts (mastitis) to cure them* |

*had any serious disease that has caused death of many people but who knows these diseases don't announce when they are coming. Their only announcement is when you wake up and people are falling sick one after another and you don't know why" (Women's FGD Kaigo)*

## Healthcare services and health seeking behavior

Although there are heath centers within the area, communities feel that the healthcare services are not adequate; drugs not enough, long waiting hours and health workers rarely ask about any possible contact with wildlife or even domestic animals. Consequently, communities' use traditional means to treat diseases when the illness is suspected to be as results of contact with wildlife. This method only offers temporary relief, leading to persistent ill health or death among those affected.

*"Some people go to health centers; others use herbs the one who provided herb for rabies died. Herbs are just what we use because we do not have alternative. Even when you go to hospital, they will just give you Panadol after staying there for the whole day. We prevent diseases by avoiding interaction or going to infected people, following advice staying clean using, mosquito nets, prevention measures found in healthy centers, observing what is necessary and you do, on radio, in workshops"*

## Quantitative results

**Summary of demographic characteristics.**　A total of 370 individuals were interviewed (response rate 96.1%, 370/385). Majority of the respondents were selected from Kiziranfumbi sub-county (51.6%). The percentage of men and women who participated in the study was 56.6% and 43.8% respectively. Education level of respondents was low, with only 18.7% having attained a secondary education and 19.5% having not attended school. Less than a half (40.8%) of those interviewed lived in permanent housing and the majority (80.8%) were married. The majority (54.1%) of respondents were 36 years of age and above. Majority of respondents were Catholics and Protestants (46.2% each) with Muslims forming a small proportion (2.2%).

## Human-wildlife interaction

Respondents reported increased interaction with wildlife at the forest edge (n = 179, 48.5%) and maize was the crop most commonly shared among humans and wildlife 80% (n = 295). Almost half of the respondents (n = 162, 43.8%) reported sorting and using leftovers from wildlife as food. Respondents reported staying in houses with poor ventilation (n = 174, 47.0%) and most people (n = 317, 88.1%) had bats and rats in their houses. Also, 33% (n = 122) of the respondents used open water sources that they shared with wildlife. The majority of respondents (n = 302, 81.6%) neither treated nor boiled water before drinking.

**Drivers of hunting and health risk perception.**　Overall, respondents participated in hunting and or consumption of wildlife. Majority of respondents (64.7%, CI: 59.6–69.6) agreed that hunted meat is shared with the community, while 29% (CI: 24.4–33.9) of the respondents' hunted non-human primates such as monkeys and baboons, 45.8% (CI: 40.6–51.0) hunted (edible) rodents. According to 55.3% (CI: 50.1–60.4) of the respondents, search for animal protein is the main reason why communities hunt, followed by cultural practices and medicinal use (22.7%, CI: 18.6–27.4). Consumption of bush-meat is practiced and 7.9% (CI: 5.3–11.1) of the respondents had a self-reported history of falling sick after bush meat consumption; 10.9% (CI: 6.3–17.4) and 68.6% (CI: 60.1–76.2) of respondents reported occurrence of illness within

the past year and last month respectively. About five percent (5.3% CI: 2.60–9.60) of respondents who fell sick reported that some family members also showed similar signs at that time. Generally, respondents were less aware (37.0%, CI: 32.1–42.2) of diseases transmissible from wildlife to humans even though majority (88.7%, CI: 85.0–92.0) had heard of Ebola or Marburg (see Table 2)

**Factors associated with falling sick after eating Bushmeat among communities around forests fragments of Hoima.**   The factors associated with falling sick after eating Bushmeat among communities around forests fragments of Hoima were analyzed using binary logistic regression model both at bivariate and multivariable levels. Bivariate results in Table 3 indicate that compared to hunting non-human primates, there were reduced odds of falling sick after eating bushmeat among the respondents who hunted edible rats (cane rats) (cOR = 0.4, 95% CI = 0.1–0.9). Having more than seven people involved in preparation of hunted animals was associated with increased odds of falling sick after eating a wild animal (cOR = 3.6, 95% CI = 1.3–9.7) as shown in Table 3

However, none of the factors was independently associated with falling sick after consumption of bushmeat in the multivariate analysis after adjusting for confounding. This underscores the role of multiple factors leading to human illness after consumption of bushmeat and a summary of potential factors has been provided in Fig 2.

## Discussion

This study focused on understanding and describing the nature of interaction between humans and wildlife around the forest fragments of Hoima and its potential for zoonotic disease spillover to humans. Additionally, the study also explored whether humans understand the health risk associated with this constant interaction with wildlife. Results from this study reveal increased community sharing of resources (crops, fruits and water) with wildlife driven by survival needs of the people and wildlife alike as their habitats range continue to shrink due to human encroachment. Studies have highlighted that as the forest fragments become farmland, there has been a dietary change of wildlife to agriculture crops further increasing conflict between human and wildlife [14,26].

Respondents talked about persistent behaviors like hunting, bushmeat consumption and the use of wild animals' parts as long time practices. These practices have been adopted in part to cope with challenges in life but there is also a strong sense of pride, cultural and community attachment to these practices despite the associated health risks. Studies have also shown that poor rural communities have been forced into circumstances of high human-wildlife contact due to poverty and a struggle to survive, exposing them to diseases [6]. Other studies have showed that despite communities' awareness about zoonosis, hunting communities for example, still have a high preference for bushmeat [27].

Important to note is that there is a higher likelihood of acquiring wildlife related diseases when seven or more people are involved as compared to less people probably because of the time spend on the carcass. Studies have shown exposure to blood or other secretions during hunting and butchering of bushmeat or through bites and scratches from wild animals are considered a primary risk factor for a broad spectrum of other zoonotic disease transmission to humans [28,29,30]. Additionally, Studies have shown that other than ethnic reasons, hunting is done during grazing and in the absence of food predisposing communities to infectious pathogens [24,30].

Respondents from studied communities had limited understanding of the risk of zoonotic disease transmission from wildlife to humans. For example, 63% of respondents did not understand that Ebola or Marburg is associated with people coming into close contact with

**Table 2. Percent of study participants falling sick after consumption of bushmeat among communities at human-wildlife interface in Hoima March to M ay 2017.**

| Characteristic(s) | Response | Location | | | | | | Chi-Sq.(P-Value) |
|---|---|---|---|---|---|---|---|---|
| | | Kitoba (N) | % (95% CI) | Kiziranfumbi (N) | % (CI) | Overall (N) | % (CI) | |
| Fall sick | Yes | 10 | 34.5 (19.6–53.2) | 19 | 65.5 (46.8–80.4) | 29 | 7.9 (5.3–11.1) | 2.48 (0.289) |
| | No | 169 | 49.7 (44.4–55.0) | 171 | 50.3 (45.0–55.6) | 340 | 92.1 (88.9–94.7) | |
| Duration of occurrence | Years back | 5 | 33.3 (14.6–59.5) | 10 | 66.7 (40.5–85.4) | 15 | 10.9 (6.3–17.4) | **30.65 (0.0001)** |
| | Months back | 63 | 67.0 (56.9–75.8) | 31 | 33.0 (24.2–43.1) | 94 | 68.6 (60.1–76.2) | |
| | Can't recall | 10 | 10.7 (3.5–28.5) | 25 | 89.3 (71.5–96.5) | 28 | 20.4 (14.0–28.2) | |
| Similar signs in some family members | Yes | 1 | 10 (1.4–46.9) | 9 | 90 (53.1–98.6) | 10 | 5.3 (2.60–9.60) | **10.35 (0.006)** |
| | No | 78 | 44.1 (36.9–51.5) | 99 | 55.9 (48.5–63.1) | 177 | 94.7 (90.4–97.4) | |
| Aware of wildlife diseases transmissible to humans | Yes | 59 | 43.1 (35.0–51.5) | 78 | 56.9 (48.5–65.0) | 137 | 37 (32.1–42.2) | **7.59 (0.022)** |
| | No | 118 | 53.2 (46.6–59.6) | 104 | 46.8 (40.4–53.4) | 222 | 60 (54.8–65.0) | |
| | Unknown | 2 | 18.2 (4.6–50.8) | 9 | 81.8 (49.2–95.4) | 11 | 3 (1.5–5.3) | |
| Heard of Ebola and or Marburg | Yes | 155 | 47.3 (41.9–52.7) | 173 | 52.7 (47.3–58.1) | 328 | 88.7 (85.0–92.0) | 3.29 (0.193) |
| | No | 23 | 60.5 (44.4–74.7) | 15 | 39.5 (25.3–55.6) | 38 | 10.3 (7.4–13.8) | |
| | Unknown | 1 | 25.0 (3.3–76.4) | 3 | 75.0 (23.6–96.7) | 4 | 1.1 (0.3–2.7) | |
| Areas where interaction with wildlife is more common | Forest edge | 75 | 41.9 (3.5–49.3) | 104 | 58.1 (50.7–65.1) | 179 | 48.5 (44.3–53.7) | **13.416 (0.004)** |
| | Forest interior | 65 | 61.3 (51.7–70.1) | 41 | 38.7 (29.9–48.2) | 106 | 28.7 (24.2–33.6) | |
| | Surrounding bush around household | 37 | 50.0 (38.8–61.2) | 37 | 50 (38.8–61.2) | 74 | 20.1 (16.1–24.5) | |
| | Not recorded | 2 | 20.0 (5.0–54.2) | 8 | 80 (45.8–95.0) | 10 | 2.7 (1.3–4.9) | |
| Human grown food shared with wild animals | Maize | 147 | 49.8 (44.1–55.5) | 148 | 50.2 (44.5–55.9) | 80 | 80.0 (75.5–83.9) | **9.51(0.050)** |
| | Sugarcane | 12 | 70.6 (45.7–87.2) | 5 | 29.4 (12.8–54.3) | 4.6 | 4.6 (2.7–7.3) | |
| | Cassava | 8 | 33.3 (17.6–53.9) | 16 | 66.7 (46.0–82.4) | 6.5 | 6.5 (4.2–9.5) | |
| | Others | 7 | 29.2 (14.5–49.9) | 17 | 70.8 (50.1–85.5) | 6.5 | 6.5 (4.2–9.5) | |
| | None | 5 | 55.6 (25.0–82.4) | 4 | 44.4 (17.6–74.9) | 2.5 | 2.4 (1.1–4.6) | |
| How are left over from wild animals treated | Chopped and buried | 5 | 17.2 (7.3–35.40) | 24 | 82.7 (64.6–92.7) | 37 | 10.0 (7.2–13.6) | **17.0 (0.002)** |
| | Sort and use as food | 82 | 50.6 (42.9–58.3) | 80 | 49.4 (41.7–57.1) | 162 | 43.9 (38.8–49.1) | |
| | Leave them to rot in the garden | 80 | 55.6 (47.3–63.5) | 64 | 44.4 (36.5–52.7) | 144 | 39.0 (34.0–44.2) | |
| | Others | 9 | 34.6 (19.5–54.3) | 17 | 65.4 (45.6–80.9) | 26 | 7.0 (4.6–10.2) | |
| | Not recorded | 3 | 37.5 (12.5–71.6) | 6 | 62.5 (28.4–87.5) | 9 | 2.4 (1.1–4.6) | |
| Has any member of family ever been attacked by wild animals | Yes | 27 | 48.9 (43.3–54.5) | 29 | 51.1 (45.5–56.7) | 56 | 15.1 (11.6–19.2) | 1.13(0.57) |
| | No | 150 | 48.2 (35.5–61.20 | 157 | 51.8 (38.8–64.5) | 307 | 83.0 (78.8–86.7) | |
| | Non response | 2 | 28.6 (7.2–67.5) | 5 | 71.4 (32.5–92.8) | 7 | 1.9 (0.8–3.9) | |
| Is hunted meat shared in the community | Yes | 125 | 53.0 (46.6–59.3) | 111 | 47.0 (40.7–53.4) | 236 | 64.7 (59.6–69.6) | 5.61(0.061) |
| | No | 53 | 41.1 (32.9–49.8) | 76 | 58.9 (50.2–67.1) | 129 | 35.0 (30.1–40.1) | |
| | Non response | 1 | 25.0 (3.3–76.4) | 3 | 75.0 (23.6–96.7) | 4 | 1.1 (0.3–2.7) | |
| Most frequently hunted animals | Non-human primates | 48 | 44.9 (35.7–54.4) | 59 | 55.1 (45.6–64.3) | 107 | 29.0 (24.4–33.9) | 9.09(0.059) |
| | Edible rodents | 80 | 47.3 (39.9–54.9) | 89 | 52.7 (45.1–60.1) | 169 | 45.8 (40.6–51.0) | |
| | Wild ruminants | 33 | 67.3 (53.1–78.9) | 16 | 32.6 (21.0–46.9) | 49 | 13.3 (10.0–17.2) | |
| | Others | 1 | 25 (3.3–76.3) | 3 | 75.0 (23.6–96.7) | 4 | 1.1 (0.3–2.7) | |
| | None response | 17 | 42.5 (28.3–58.1) | 23 | 57.5 (41.9–71.7) | 40 | 10.8 (7.8–14.5) | |
| Reason for hunting | Cultural/Medical purpose | 47 | 55.9 (45.2–66.2) | 37 | 44.0 (33.8–54.8) | 84 | 22.7 (18.6–27.4) | **17.08(0.001)** |
| | Protein source | 106 | 51.9 (45.1–58.7) | 98 | 48.1 (41.2–54.9) | 204 | 55.3 (50.1–60.4) | |
| | Others | 6 | 17.1 (7.9–33.3) | 29 | 82.9 (66.7–92.1) | 35 | 9.5 (6.7–12.9) | |
| | Non response | 20 | 43.5 (30.0–58.0) | 26 | 56.5 (42.0–70.0) | 46 | 12.5 (9.3–16.3) | |

(*Continued*)

**Table 2.** (Continued)

| Characteristic(s) | Response | Location | | | | | | Chi-Sq.(P-Value) |
|---|---|---|---|---|---|---|---|---|
| | | Kitoba (N) | % (95% CI) | Kiziranfumbi (N) | % (CI) | Overall (N) | % (CI) | |
| Number of people involved in preparation of hunted animals | 1–3 People | 83 | 63.8 (55.2–71.7) | 47 | 36.2 (28.3–44.8) | 130 | 35.2 (30.4–40.3) | **19.71(0.001)** |
| | 4–7 People | 37 | 44.0 (33.8–54.8) | 47 | 56.0 (45.2–66.2) | 84 | 22.8 (18.6–27.4) | |
| | Above 7 people | 33 | 37.5 (28.0–48.1) | 55 | 62.5 (51.9–72.0) | 88 | 23.8 (19.6–28.5) | |
| | Non response | 26 | 38.8 (29.7–50.9) | 41 | 61.2 (49.1–72.1) | 67 | 18.2 (14.3–22.5) | |

wildlife like Chimpanzees. Additionally, 43.8% of respondents reported regular consumption of leftovers from wildlife crop raids as food. Communities reported adapting to eating other wild animals (bushmeat) that they never used to eat before like squirrels because of the profound belief that bushmeat is medicinal. Interaction with rodents, bats and other wildlife are a long-standing feature of life in these landscapes and rodents are routinely hunted and consumed but also found in gardens of most respondents. An earlier study done in Hoima, recommends alternative viable livelihood and educational outreaches as means to promote co-existence between human and wildlife [31]. Previous studies have linked this interaction to human encroachment into and modification of wildlife habitats due to population increase and competing humans needs [32].

**Table 3. Factors associated with falling sick after eating Bushmeat among communities around forests fragments of Hoima March to May 2017.**

| Characteristic(s) | | Crude analysis | | | Adjusted analysis | | |
|---|---|---|---|---|---|---|---|
| | Categories | cOR | 95% CI | P-value | aOR | 95% CI | P-value |
| Sex | Female | 1 | | | | | |
| | Male | 1.3 | 0.6–2.9 | 0.49 | | | |
| Age Groups | <25 years | 1 | | | 1 | | |
| | 25–35 years | 1.9 | 0.4–9.4 | 0.419 | 2.1 | 0.4–11.0 | 0.377 |
| | 36+ years | 2.7 | 0.6–11.8 | 0.197 | 3 | 0.6–14.2 | 0.171 |
| Household Size | 1–3 | 1 | | | 1 | | |
| | 4–7 | 0.5 | 0.2–1.3 | 0.156 | 0.5 | 0.2–1.2 | 0.124 |
| | 8+ | 0.7 | 0.2–2.1 | 0.525 | 0.6 | 0.2–1.9 | 0.359 |
| Know of any diseases transmitted by wild animals to human | Yes | 1 | | | | | |
| | No | 0.6 | 0.3–1.3 | 0.222 | | | |
| | Non-response | 1 | | | | | |
| Ever heard of Ebola or Marburg | Yes | 1 | | | | | |
| | No | 0.6 | 0.1–2.7 | 0.532 | | | |
| Mostly frequently hunted animals | Non-human primates | 1 | | | 1 | | |
| | Edible rodents | 0.4 | 0.1–0.9 | 0.029 | 0.5 | 0.2–1.4 | 0.183 |
| | Wild ruminants | 0.7 | 0.2–2.1 | 0.474 | 1.1 | 0.3–4.4 | 0.92 |
| | Others | 3.6 | 0.3–42.3 | 0.312 | 3.1 | 0.2–44.6 | 0.406 |
| | Non-response | 0.6 | 0.2–2.4 | 0.493 | 0.5 | 0.1–3.0 | 0.475 |
| Number of people involved in preparation of hunted animals | 1–3 People | 1 | | | | | |
| | 4–7 People | 0.8 | 0.2–3.2 | 0.729 | 0.7 | 0.2–3.3 | 0.697 |
| | Above 7 People | 3.6 | 1.3–9.7 | 0.014 | 2.9 | 0.8–9.7 | 0.091 |

cOR = Crude Odds Ratio; aOR = Adjusted Odds Ratio, CI = Confidence interval (at 95%)

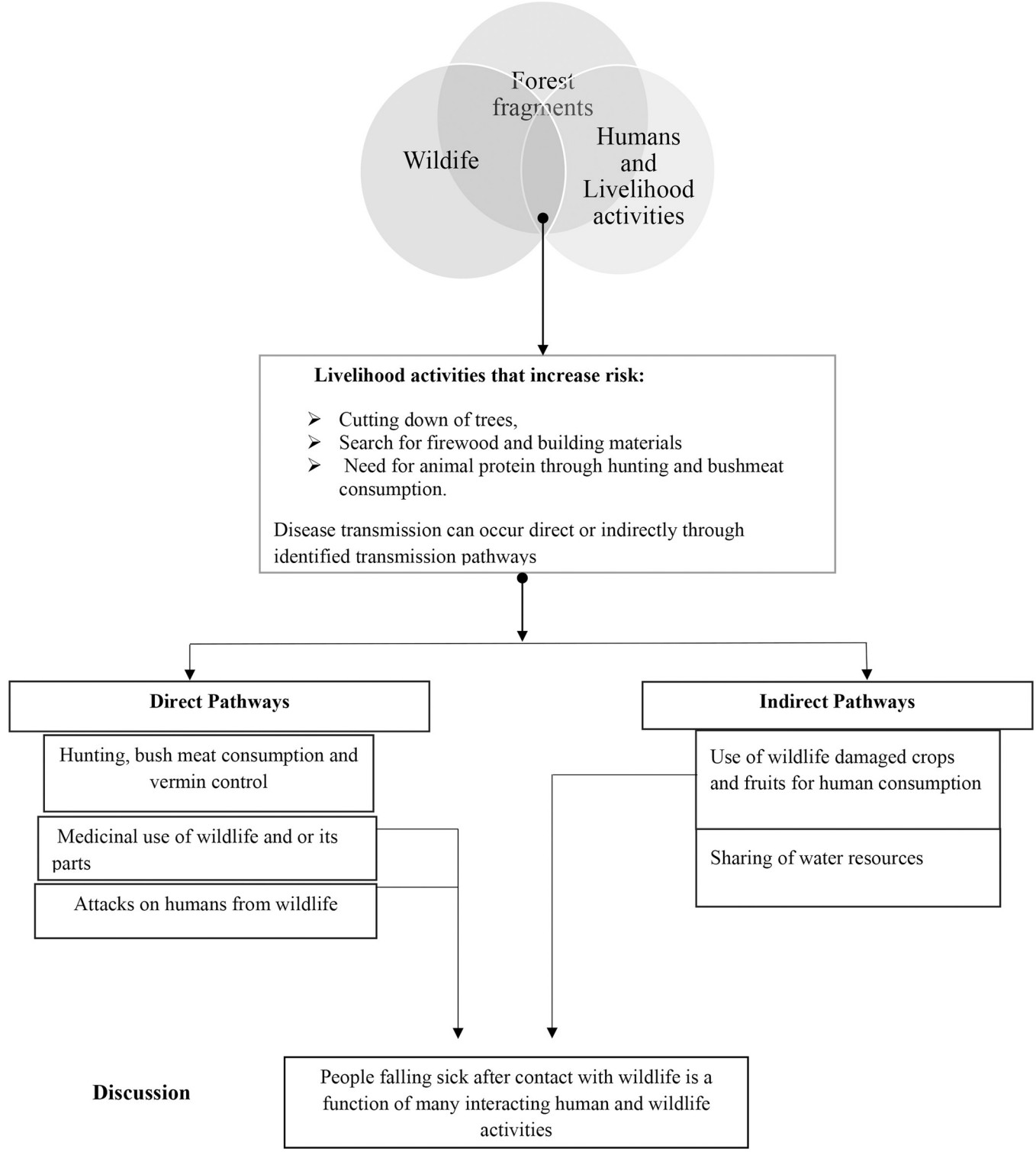

**Fig 2. Summary of potential Zoonotic Pathways at the human-wildlife interface Hoima Forest Fragments.**

Communities acknowledge that clinical care and even diagnostic investigations are poor and most cases of ill health will resolve on their own. Research has shown that most zoonotic diseases remain neglected because of the lack of adequate diagnostic laboratory services and the fragmented collection of data that cannot be used to inform any recommendations [18]. Findings from a study in Uganda shows that health workers in zoonosis hotspots have low knowledge on zoonotic diseases, a factor that makes early detection a challenge especially at the community level [29,33]. Similarly, evidence from the West Africa Ebola outbreak shows that although the risks to human infections from animals remains a threat, the population and health services in many developing countries remain unprepared for the next outbreak [34] yet the cost of managing such infectious disease outbreaks is greater than the cost of avoiding them [35]. Furthermore, prediction studies have shown that about 97% of 22 million people in rural Africa live in areas suitable for zoonotic transmission of disease like Ebola, and Uganda ranks high among countries that have recorded such outbreaks [36].

Overall, findings suggest that knowledge about zoonoses is not adequately filtering down to the communities to impact human behaviors regarding wildlife interaction, despite awareness about the existence of specific zoonoses like Ebola. This is a reflection of the international and national focus and investment in certain diseases of public health concern and the neglect of endemic zoonotic diseases that significantly contribute to the disease burden in these communities. This also has implications for zoonotic disease spread and highlights the extent to which education and dissemination of information can mitigate future outbreaks of such zoonotic diseases [37].

## Conclusion

The results from this study suggest that there is interaction between humans and wildlife among communities at the human wildlife interface in Hoima. This interaction is largely driven by human needs and creates a potential threat for disease spillover to humans due to persistent hunting-bushmeat activities. This is further complicated by communities' lack of awareness of the health risk associated with close wildlife interaction and inadequate health care services. This therefore calls for concerted efforts among government agencies and partners to work with the communities and create adequate awareness about zoonotic diseases of wildlife origin. These findings will be shared with the local government of Hoima and at different scientific meetings to highlight some of the key cultural issues defining the risk of disease transmission along the human-wildlife interface in an area of high biodiversity and potential strategies to address this risk.

## Acknowledgments

We are grateful to Conservation and Ecosystem Health Alliance and Hoima district local government for their support towards field work.

## Author Contributions

**Conceptualization:** Shamilah Namusisi, Michael Mahero, Dominic Travis, Katherine Pelican, Cheryl Robertson.

**Data curation:** Shamilah Namusisi.

**Formal analysis:** Shamilah Namusisi, Michael Mahero, Cheryl Robertson.

**Funding acquisition:** Shamilah Namusisi, Dominic Travis, Katherine Pelican.

**Investigation:** Shamilah Namusisi, Lawrence Mugisha.

**Methodology:** Shamilah Namusisi, Michael Mahero, Dominic Travis, Katherine Pelican, Cheryl Robertson, Lawrence Mugisha.

**Project administration:** Shamilah Namusisi, Dominic Travis, Lawrence Mugisha.

**Resources:** Dominic Travis, Katherine Pelican.

**Supervision:** Dominic Travis, Katherine Pelican, Cheryl Robertson.

**Validation:** Shamilah Namusisi, Michael Mahero, Dominic Travis, Lawrence Mugisha.

**Visualization:** Shamilah Namusisi, Dominic Travis, Lawrence Mugisha.

**Writing – original draft:** Shamilah Namusisi.

**Writing – review & editing:** Shamilah Namusisi, Michael Mahero, Dominic Travis, Katherine Pelican, Cheryl Robertson, Lawrence Mugisha.

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
