## [Decision Letter · Decision Letter 0]

7 Oct 2019

Dear Dr. Namusisi:

Thank you very much for submitting your manuscript "Zoonotic disease risk in South-Western Uganda: characterizing the human-wildlife interface in a biodiversity hot spot" (#PNTD-D-19-01184) for review by PLOS Neglected Tropical Diseases. Your manuscript was fully evaluated at the editorial level and by independent peer reviewers. The reviewers appreciated the attention to an important problem, but raised some substantial concerns about the manuscript as it currently stands. These issues must be addressed before we would be willing to consider a revised version of your study. We cannot, of course, promise publication at that time.

We therefore ask you to modify the manuscript according to the review recommendations before we can consider your manuscript for acceptance. Your revisions should address the specific points made by each reviewer. 

When you are ready to resubmit, please be prepared to upload the following:

(1) A letter containing a detailed list of your responses to the review comments and a description of the changes you have made in the manuscript.

(2) Two versions of the manuscript: one with either highlights or tracked changes denoting where the text has been changed (uploaded as a "Revised Article with Changes Highlighted" file); the other a clean version (uploaded as the article file).

(3) If available, a striking still image (a new image if one is available or an existing one from within your manuscript). If your manuscript is accepted for publication, this image may be featured on our website. Images should ideally be high resolution, eye-catching, single panel images; where one is available, please use 'add file' at the time of resubmission and select 'striking image' as the file type. 

Please provide a short caption, including credits, uploaded as a separate "Other" file. If your image is from someone other than yourself, please ensure that the artist has read and agreed to the terms and conditions of the Creative Commons Attribution License at http://journals.plos.org/plosntds/s/content-license (NOTE: we cannot publish copyrighted images). 

(4) If applicable, we encourage you to add a list of accession numbers/ID numbers for genes and proteins mentioned in the text (these should be listed as a paragraph at the end of the manuscript). You can supply accession numbers for any database, so long as the database is publicly accessible and stable. Examples include LocusLink and SwissProt.

(5) To enhance the reproducibility of your results, we recommend that you deposit your laboratory protocols in protocols.io, where a protocol can be assigned its own identifier (DOI) such that it can be cited independently in the future. For instructions see http://journals.plos.org/plosntds/s/submission-guidelines#loc-methods

While revising your submission, please upload your figure files to the Preflight Analysis and Conversion Engine (PACE) digital diagnostic tool, https://pacev2.apexcovantage.com/ PACE helps ensure that figures meet PLOS requirements. To use PACE, you must first register as a user. Then, login and navigate to the UPLOAD tab, where you will find detailed instructions on how to use the tool. If you encounter any issues or have any questions when using PACE, please email us at figures@plos.org.

We hope to receive your revised manuscript by Dec 06 2019 11:59PM. If you anticipate any delay in its return, we ask that you let us know the expected resubmission date by replying to this email.

To submit a revision, go to https://www.editorialmanager.com/pntd/ and log in as an Author. You will see a menu item call Submission Needing Revision. You will find your submission record there. 

Sincerely,

Charles Apperson

Guest Editor

Bruce Lee

Deputy Editor

Reviewer's Responses to Questions

**Key Review Criteria Required for Acceptance?**

**Methods**

-Are the objectives of the study clearly articulated with a clear testable hypothesis stated?

-Is the study design appropriate to address the stated objectives?

-Is the population clearly described and appropriate for the hypothesis being tested?

-Is the sample size sufficient to ensure adequate power to address the hypothesis being tested?

-Were correct statistical analysis used to support conclusions?

-Are there concerns about ethical or regulatory requirements being met?

Reviewer #1: See comments above on summary section. The choice of qualitative and quantitative seems appropriate for the objective of the study, but overall methods are insufficiently explained.

Reviewer #2: (No Response)

Reviewer #3: Field methods may be appropriate. See additional comments below.

Study design is fine for what they wish to report.

Characterizing the actual disease risk w/o a known response (specific disease) that can be attributable to zoonoses is not possible in this study. There is no pathogen or serological testing and reported disease is limited. Recommend revising title.

**Results**

-Does the analysis presented match the analysis plan?

-Are the results clearly and completely presented?

-Are the figures (Tables, Images) of sufficient quality for clarity?

Reviewer #1: See comments on summary section. Some analysis are mentioned but not reported in results.

Reviewer #2: (No Response)

Reviewer #3: See additional notes re: interpreting 95% CI for ORs

Tables will need editorial assistance. 

Need 95% CI for proportions where applicable.

**Conclusions**

-Are the conclusions supported by the data presented?

-Are the limitations of analysis clearly described?

-Do the authors discuss how these data can be helpful to advance our understanding of the topic under study?

-Is public health relevance addressed?

Reviewer #1: Adequate

Reviewer #2: (No Response)

Reviewer #3: See additional notes re: interpreting 95% CI for ORs

**Editorial and Data Presentation Modifications?**

Reviewer #1: (No Response)

Reviewer #2: (No Response)

Reviewer #3: See summary and general comments

**Summary and General Comments**

Reviewer #1: The paper provides an overview of the nature of interaction between humans and wildlife around the forest fragments of Hoima in Uganda, as well as the level of knowledge of the community regarding the risk associated with wildlife interactions. Although the study is not particularly novel or unique, it provides an interesting summary of the challenges and types of interactions in the human-wildlife interface. Some sections of the papers need to be heavily edited and clarified. For example, the introduction needs to be written. It was numerous sentences that are unclear or repetitive. Similarly, the section on quantitative analysis is insufficient to understand what was done. It seems like the models were univariate, but multivariate methods are mentioned on results, but not described. 

Specific comments:

L69-71: The sentence is unclear, are 60 % of the infectious diseases zoonotic in origin?

L78-89: Rephrase the sentence, e.g. “The burden of infectious diseases is particularly high in Sub-Saharan Africa. In addition, poor communities are disproportionally affected by climate and environmental changes that further drive the emergence of infectious diseases. “

L85: Spell out EBOV and DRC

L88: Missing reference

L88-L90: Rephrase: “forest fragments are inhabited by a mobile population of about 5000 chimpanzees that move within and between forest-farm habitats, causing increased conflict with human farming communities”

L92-L101: Sentences are duplicated.

L388: “Having more than seven people involved in preparation of hunted animals had a higher risk associated with sickness (cOR=3.6, 95% CI=1.3-9.7) compared to involving 1-3 people” this is the only significant cOR. It should be discussed. 

L140-41: Are the number of villages and forest fragment equally represented, how are both related (1 to 1)?

L179: Were 10 FGD per village? That would give you 80m which then it should give you more than the 370 individuals indicated above. 

L211-220: Description of the logistic models is inadequate. 

L233: what kind of risks are perceived by the communities

Table 1 should also summarize the most frequent hunted animals and all uses. 

Recall bias is going to play a role in this study

Where are the multivariate results? 

Discussion is well-written but overall it provides little reference back to the results.

Reviewer #2: (No Response)

Reviewer #3: This study is a cross-sectional, mixed methods study based on a “face to face” administered (semi-structured) questionnaire about zoonotic disease and risk behaviors in Uganda. This study has some merit, but the manuscript needs additional editorial work and some major revisions by the authors.

Oxford comma, line 34 (After Minnesota)

Statements (Line 36-38) read as if they are presented as facts. They are participant impressions. There should be clear language stating such –or perhaps the judicious use of quotes. 

In abstract, and throughout manuscript, please include confidence intervals for proportions/percentages so that the reader can infer certainty based on sample size.

Line 44: Sentence needs revision. 

Line 46: “fora” changed to “for a”

Line 47: missing word (? campaigns) after education

Line 59: “less aware” suggests a comparison … consider other word choice

Line 63: “sensitized” should be defined here or consider other word choice

Line 64: Consider removing “Hoima” here and add “agencies” or “officials” after government

Line 78: Sentence needs revision (? Edit as “…is considered to have the highest burden…”)

Line 85: EBOV should be defined [Review author guidelines re: abbreviations]

Line 88: Missing reference

Line 140: Remove extra “.”

Line 157: How were household randomized?

Line 161: Change “10focus” to “10 focus”

Line 161: Add space after “(FGDs)”

Line 170: Consider “We used the services of a local…”

Line 184: add comma after “humans”

Line 185: Was consent obtained?

Line 195 and throughout: Does “Runyoro” need to be italicized 

Line 200: Please expand on analysis methods for qualitative data analysis

Line 202: Consider “Validity” en lieu of “Trustworthiness”

Note: the manuscript needs editorial assistance. Missing articles and typographical errors abound in this draft. I am not making additional basic edits/suggestions beyond line 202.

Data analysis: An odds ratio is not the same as a relative risk. The statements that include “times at risk” need to be revised (or the data analysis should include relative risk). Furthermore, if the 95% CI contains 1, then it isn’t significant. Thus, most of the reported OR in lines 327-339 are NOT significant. Thus, inference should be limited as it isn’t supported by data.

Data tables and images are mixed within the manuscript text. Recommend that authors conform to standard author guidelines. Manuscript sections and headings are not standardized. Review guidelines. 

Conclusions state that there is increasing interaction between humans and wildlife based on the study. No clear evidence was provided for this. The discussion and conclusions need significant editorial assistance. 

References are not in standard format. Need to revise many to the standard format.

PLOS authors have the option to publish the peer review history of their article (what does this mean?). If published, this will include your full peer review and any attached files.

Reviewer #1: No

Reviewer #2: No

Reviewer #3: No

---

## [Editor Report · Decision Letter 1]

27 Feb 2020

Dear Dr. Namusisi: We have received your revised manuscript, including your responses to the reviewers' comments and concerns. The reviewers recommended that your manuscript required a major revision before it would be suitable for publication. They made numerous comments and provided extensive suggestions for improving your manuscript. We find that your response to these comments and concerns is not adequate. Accordingly, we are requesting that you address each review separately and provide a response to each point raised by each reviewer. Once we have received a more comprehensive response to the reviews, we will send your revised manuscript out for review.

Thank you for your patience in the process and for your interest in publishing in PLoS Neglected Tropical Diseases. 

Sincerely yours,

Charles S. Apperson

Guest Editor

We cannot make any decision about publication until we have seen the revised manuscript and your response to the reviewers' comments. Your revised manuscript is also likely to be sent to reviewers for further evaluation.

Sincerely,

Charles Apperson

Guest Editor

Bruce Lee

Deputy Editor

Dear Dr. Namusisi: We have received your revised manuscript, including your responses to the reviewers' comments and concerns. The reviewers recommended that your manuscript required a major revision before it would be suitable for publication. They made numerous comments and provided extensive suggestions for improving your manuscript. We find that your response to these comments and concerns is not adequate. Accordingly, we are requesting that you address each review separately and provide a response to each point raised by each reviewer. Once we have received a more comprehensive response to the reviews, we will send your revised manuscript out for review.

Thank you for your patience in the process and for your interest in publishing in PLoS Neglected Tropical Diseases. 

Sincerely yours,

Charles S. Apperson

Guest Editor
---

## [Decision Letter · Decision Letter 2]

18 Jun 2020

Dear Dr. Namusisi,

Thank you very much for submitting your manuscript "A descriptive study of zoonotic disease risk at the human-wildlife interface in a biodiversity hot spot in South Western Uganda" for consideration at PLOS Neglected Tropical Diseases. As with all papers reviewed by the journal, your manuscript was reviewed by members of the editorial board and by several independent reviewers. The reviewers appreciated the attention to an important topic. Based on the reviews, we are likely to accept this manuscript for publication, providing that you modify the manuscript according to the review recommendations. 

Sincerely,

Charles Apperson

Guest Editor

Bruce Lee

Deputy Editor

Reviewer's Responses to Questions

**Key Review Criteria Required for Acceptance?**

**Methods**

-Are the objectives of the study clearly articulated with a clear testable hypothesis stated?

-Is the study design appropriate to address the stated objectives?

-Is the population clearly described and appropriate for the hypothesis being tested?

-Is the sample size sufficient to ensure adequate power to address the hypothesis being tested?

-Were correct statistical analysis used to support conclusions?

-Are there concerns about ethical or regulatory requirements being met?

Reviewer #1: In general methods are appropriate and the population clearly defined.

Reviewer #2: (No Response)

Reviewer #3: Objectives are generally clearly described. 

Recommend (line 230) additional description of what "data cleaned' means.

Sample sizes are impressive and likely provide sufficient power to address comparative questions.

I do have some concerns over the interpretations of the odds ratios. (see comments in results/conclusions critiques below)

**Results**

-Does the analysis presented match the analysis plan?

-Are the results clearly and completely presented?

-Are the figures (Tables, Images) of sufficient quality for clarity?

Reviewer #1: Yes, mostly negative results

Reviewer #2: (No Response)

Reviewer #3: Some formatting issues (likely PDF) for figure 2. Other figures/tables are clear. line 373 references table 4, but should be table 3 (no table 4 is provided)

The cORs are not appropriately interpreted with the given 95%CI. For example (Line 373) the authors stated that men are 1.3 times more likely to get sick: since this is an odds ratio, that is an incorrect interpretation -- ORs are not the same as risk ratios. In addition, in multiple places, OR data is presented with accompanying 95% CIs that have an interval that includes "1" -- thus the authors should not conclude there is a difference. These issues must be addressed before publication

**Conclusions**

-Are the conclusions supported by the data presented?

-Are the limitations of analysis clearly described?

-Do the authors discuss how these data can be helpful to advance our understanding of the topic under study?

-Is public health relevance addressed?

Reviewer #1: Limited conclusions given the lack of conclusive results.

Reviewer #2: (No Response)

Reviewer #3: Recommend reviewing how to interpret ORs: https://www.ncbi.nlm.nih.gov/pmc/articles/PMC5253299/

**Editorial and Data Presentation Modifications?**

Reviewer #1: L27: remove comma after district

L125: delete ‘of’ and replace ‘has’ for ‘have’

L127: Full stop after humans. Delete for example and edit the next sentence as follows: “Degradation results in higher contact within existing habitat and causes significant migration of wildlife out of the degraded environments into human settlements.”

L129: What activities? Please specify.

L132: Finish the list (a), b)…).

L215: Rephrase sentence: “The translator transcribed recordings to English. 

L218: Delete that

L241: Do you mean p-value less than 0.05?

Table 2, provide label for the last CI

L440-L441: Unclear what the point of the sentence is.

Reviewer #2: (No Response)

Reviewer #3: Still some typos, but the manuscript is greatly improved. The cOR interpretation must be edited for clarity and statistical accuracy.

**Summary and General Comments**

Reviewer #1: The study is mostly descriptive in nature with negative results, but this version is improved and addresses most previous comments reviewer.

Reviewer #2: The authors provide a much-improved version of the manuscript. I appreciate the attention to my previous major comments, which I think have increased the clarity of the work. I do have three other additional major comments, as well as a series of minor/grammatical comments, which I think will help convey the message more clearly. In addition, I encourage the authors to correct punctuation issues such as inconsistent/multiple spacing and use/exclusion of commas, which I have not commented on specifically.

Reviewer #3: The authors have a solid study that is original and adds to the body of knowledge about spillover risk in Uganda based on cultural practices and place. The work is important and has relevance beyond an academic readership -- thus should be of interest to practitioners in multiple fields. Field methods appear sound although I seek clarity about what "data cleaning" actually means. Statistical methods are appropriate, but need some work with interpreting cORs.

PLOS authors have the option to publish the peer review history of their article (what does this mean?). If published, this will include your full peer review and any attached files.

Reviewer #1: No

Reviewer #2: No

Reviewer #3: No
---

## [Editor Report · Decision Letter 3]

23 Jul 2020

Dear Dr. Namusisi,

We are pleased to inform you that your manuscript 'A descriptive study of zoonotic disease risk at the human-wildlife interface in a biodiversity hot spot in South Western Uganda' has been provisionally accepted for publication in PLOS Neglected Tropical Diseases.

Best regards,

Charles Apperson

Guest Editor

Bruce Lee

Deputy Editor

---

## [Editor Report · Acceptance letter]

31 Dec 2020

Dear Dr. Namusisi,

We are delighted to inform you that your manuscript, "A descriptive study of zoonotic disease risk at the human-wildlife interface in a biodiversity hot spot in South Western Uganda," has been formally accepted for publication in PLOS Neglected Tropical Diseases.

Best regards,

Shaden Kamhawi

co-Editor-in-Chief

Paul Brindley

co-Editor-in-Chief
